# Hierarchical Macroporous PolyDCPD Composites from Surface-Modified Calcite-Stabilized High Internal Phase Emulsions

**DOI:** 10.3390/polym15010228

**Published:** 2023-01-01

**Authors:** Ali Eslek, Hatice Hande Mert, Meltem Sözbir, Mohamed Alaasar, Emine Hilal Mert

**Affiliations:** 1Department of Polymer Materials Engineering, Institute of Graduate Studies, Yalova University, 77200 Yalova, Türkiye; 2Department of Chemical Engineering, Faculty of Engineering, Yalova University, 77200 Yalova, Türkiye; 3Department of Chemistry, Faculty of Science, Cairo University, Giza 12613, Egypt; 4Department of Chemistry, Martin Luther University Halle-Wittenberg, Kurt Mothes Str. 2, D-06120 Halle (Saale), Germany; 5Department of Polymer Materials Engineering, Faculty of Engineering, Yalova University, 77200 Yalova, Türkiye

**Keywords:** high internal phase emulsion, macroporous polymers, polyDCPD, calcite

## Abstract

High Internal Phase Emulsions (HIPEs) of dicyclopentadiene (DCPD) were prepared using mixtures of surface-modified calcite (*m*Calcite) and a non-ionic surfactant. Twelve different emulsion formulations were created using an experimental design methodology. Three distinctive levels of the internal phase ratio, the amount of *m*Calcite loading, and the surfactant were used to prepare the HIPEs. Accordingly, macroporous polyDCPD composites were synthesized by performing ring-opening metathesis polymerization (ROMP) on the HIPEs. The variations in the morphological and physical properties of the composites were investigated in terms of experimental parameters. In the end, five different model equations were derived with a confidence level of 95%. The main and binary interaction effects of the experimental parameters on the responses, such as the average cavity size, interconnecting pore size, specific surface area, foam density, and compression modulus, were demonstrated. The synergistic interaction between the amount of surfactant, the amount of *m*Calcite loading, and the internal phase ratio appeared to have a dominant role in the average cavity diameter. The solo effect of the internal phase ratio on the interconnecting pore size, foam density, and compression modulus was confirmed. In addition, it was demonstrated that the specific surface area of the composites was mainly changed depending on the amount of *m*Calcite loading.

## 1. Introduction

Macroporous poly(High Internal Phase Emulsion) (polyHIPE) foams offer the advantages of permeable monolith structures in many fields in which transfer phenomena are important [1,2,3,4]. PolyHIPEs can serves as supports in adsorption processes [5,6], catalysis [7,8,9], tissue engineering [10,11,12], storage of gases [13] and solvents [14], thermal energy storage [15,16,17], and in the preparation of the electrode material of supercapacitors [18,19,20]. In this context, especially in recent years, the use of polyHIPEs as support materials for preparing composite phase change materials (PCMs) has become a research topic of great interest. Several groups have reported polyHIPE/PCM composites for latent heat storage applications [15,16,17,21]. Studies on polyHIPE/PCM composites have reported monolithic polyHIPEs displaying leak-tightness properties during phase changes [15,16,17,21]. This can be accomplished because of the specific open cellular structure, where cavities are connected through pore throats [2,3,4]. However, to facilitate and disseminate the use of polyHIPEs as support material in industry, improvable properties, such as the mechanical strength of the matrix [4,22] and thermal conductivity [23,24,25], should also be considered.

The mechanical strength of the polymer shell or skeleton is important in terms of the sustainability of the performance during use. Low mechanical strength can easily cause damage of the support material under load or pressure, which can lead problems in the field of application. PolyHIPEs usually exhibit low mechanical strength due to their macroporous structure [4,22,26]. This is the main drawback of polyHIPEs in industrial applications. Scientists have made several attempts to resolve this problem, including creating a polymer skeleton with high mechanical strength [22,26], reducing the total porosity [27], or using fillers [27,28,29]. In this respect, polydicyclopentadiene (polyDCPD)-based polyHIPEs have been reported in the literature to have high mechanical strength [5,22,30,31].

The incorporation of micro- or nano-sized particles into the structure of polyHIPEs can also be used to improve thermal conductivity while providing additional benefits to the resulting material in terms of functionality [2,31]. Moreover, surface-modified particles might also serve in emulsion stabilization processes, depending on the presence of suitable surface groups consisting of hydrophilic and hydrophobic parts [5,29,32]. In our previous study, we demonstrated that surface-modified cellulose nano crystals (CNCs) can be efficiently used for the stabilization of DCPD-based HIPEs and that the obtained polyHIPE/CNC nanocomposites can be used effectively in the removal of organic dyes from aqueous media [5].

In the present work, we focused on synthesizing polyDCPD composites that can be used as potential support materials for PCM applications. Since calcite has relatively high thermal conductivity, improves polymer mechanical properties, and is cost-effective, we used micronized calcite as a filler. To improve the compatibility between the calcite particles and the DCPD, we accomplished the surface modification of the calcite using a PEG–PPG–PEG triblock copolymer. Accordingly, we prepared macroporous polyDCPD composites from the ring-opening metathesis polymerization (ROMP) of HIPEs, which were stabilized using a combination of surface-modified calcite (*m*Calcite) particles and a non-ionic polymeric surfactant. Because common emulsion parameters effect the pore architecture and physical properties of polyHIPEs [33,34,35], we performed a systematical study for the synthesis of polyHIPE composites. With this aim, we used an experimental design methodology to propose an experimental route in the design of macroporous polyDCPD/*m*Calcite composites for future applications. As a result, we expressed the variations in the pore structure, specific surface area, foam density, and compression modulus of the obtained polyDCPD composites with experimental parameters using mathematical model equations. To the best of our knowledge, this is the first study to report the design of a polyDCPD composite from surface-modified calcite (*m*Calcite) particles.

## 2. Materials and Methods

### 2.1. Materials

Dicyclopentadiene containing butylated hydroxytoluene (BHT) as a stabilizer (Sigma-Aldrich, St. Louis, MS, USA), Pluronic^®^ L-121 (poly(ethylene glycol)-block-poly(propylene glycol)-block-poly(ethylene glycol), average Mn ~4400, composition: PEG, 30 wt%, hydroxyl number (mg KOG/g): 20–30, HLB: 1–7, density: 1.006 g/mL at 25 °C, Sigma-Aldrich), Pluronic^®^ P-123 (poly(ethylene glycol)-block-poly(propylene glycol)-block-poly(ethylene glycol), average Mn ~ 5800, composition: PEG, 30 wt%, feed ratio: ethylene oxide:propylene oxide:ethylene oxide (EO:PO:EO): 20:70:20, hydroxyl number (mg KOG/g): 20.0–30.0, HLB: 7–9, density: 1.018 g/mL at 25 °C, Sigma-Aldrich), pentaerythritol tetrakis (3,5-di-tert-butyl-4-hydroxyhydrocinnamate 98%, Sigma-Aldrich), Grubbs Catalyst^®^ M204 (Umicore, melting point: 143.5–148.5 °C, Sigma-Aldrich), and toluene (ACS reagent, ≥99.5%, Sigma-Aldrich) were all purchased from Sigma-Aldrich and used as received. Micronized calcite was kindly donated by Miner Madencilik (powder ≈ 40 μm, CaCO_3_ content: 80.0–99.9%, Mohs hardness: 3, color: bright white to light grey, Niğde/Türkiye) and used without applying any pre-treatment. Ethanol was technical grade and used without purification. In all experiments, ultrapure deionized water was used after degassing.

### 2.2. Methods

#### 2.2.1. Modification of Calcite

Calcite modification was carried out using the cryoscopic approach. For this purpose, 1.0 g of calcite was mixed with 100 mL of deionized water in a beaker. Next, it was placed on a magnetic stirrer and stirred for 30 min (at 40 °C and 500 rpm). Afterwards, 2.0 g of Pluronic^®^ P-123 was added into the mixture and stirring was continued for another 24 h. The mixture was then placed in a freezer and kept at −40 °C for 24 h. Finally, the mixture was dried in a laboratory freeze dryer and the modified calcite (*m*Calcite) particles were obtained.

#### 2.2.2. Synthesis of polyDCPD Composites and Evaluation Based on Statistical Analysis

Hierarchical macroporous polyDCPD composites were synthesized by ROMP of w/o type DCPD-based Pickering-HIPE templates. For this purpose, *m*Calcite particles were used for emulsion stabilization together with a non-ionic Pluronic^®^ L-121 triblock copolymer. The HIPE templates were formulated using an experimental design approach and the experimental results were statistically analyzed using Minitab^®^ 21.1© 2021 (Minitab, LLC, State College, PA, USA) statistical software for Windows. Three Basic experimental parameters (the surfactant (Pluronic^®^ L-121) amount (A), the amount of *m*Calcite loading (B), and the internal phase ratio (C)), which are decisive for the determination of the final material properties, were used in the creation of the experimental design plan. In this respect, three different levels (low (−1), medium (0), and high (+1)) were determined for each parameter, and these are presented in Table 1. The other parameters, such as monomer amount, catalyst amount, stirring speed, and duration, were kept constant during the experiments. The experimental design matrix used for the HIPE formulations (see Table 2) was created using the Minitab^®^ 21.1©2021 (Minitab, LLC) statistical software for Windows. The polyDCPD composites were prepared and the data obtained through material characterization were entered into the Minitab^®^ 21.1 software according to the experimental design matrix given in Table 2. The experimental design matrix is a table which demonstrates all the combinations of the factors at levels described as (−1) for the low level, (0) for a center point, and (1) for the high level. According to Table 2, the column of “Std Order” shows the order of the experiment. The “Center Pt” column shows experiments for which parameter levels are set in the center of the low and high settings. In this column, “1” represents the corner point while “0” denotes a center point. Moreover, the “Blocks” column indicates the groups of experiments which are conducted under the same conditions. 

With this approach, 12 different DCPD-based HIPE formulations were created, whilst formulations based on StdOrder 9, 10, 11, and 12 in Table 2 are identical. These formulations were used to synthesize the control samples.

DCPD-based HIPEs were prepared using the experimental design matrix given in Table 2. In a typical experiment, the DCPD, the *m*Calcite particles, the surfactant (Pluronic^®^ L-121), and an antioxidant (pentaerythritol tetrakis (3,5-di-tert-butyl-4-hydroxyhydrocinnamate), 0.2 wt% regarding to DCPD) were placed in a two-necked round-bottom glass reactor equipped with an overhead digital stirrer and a peristaltic pump. The reactor was then immersed in an oil bath which was fixed at 25 °C on a heater plate. The mixture was stirred (at 400 rpm) until a homogeneous mixture of the components was obtained. Next, the internal phase was added under constant stirring with the help of a peristaltic pump (with a pumping speed of 50 rpm). Once the addition of the internal phase was completed, stirring was continued for a further 1 h to obtain a homogeneous emulsion. After this, a Grubbs’ catalyst (1 mole% regarding the DCPD) was dissolved in 1 mL of toluene and quickly transferred to the resulting emulsion. After the addition of the catalyst, the emulsion was stirred for 1 min and rapidly transferred to glass containers. To achieve polymerization, the glass containers were placed in an air circulating oven with the lids closed and kept at 80 °C for 3 h. Afterwards, the polyDCPD composites were removed from the glass containers and extracted with ethanol for 24 h in a Soxhlet apparatus. They were then dried under vacuum at 25 °C until constant weighing was possible. Finally, 12 polyDCPD composite samples were obtained. These samples were designated x-KP, where x denotes the StdOrder number of the HIPE formulation in Table 2. The samples named 9-KP, 10-KP, 11-KP, and 12-KP are the control samples.

### 2.3. Characterization

The chemical structure of the obtained modified particles was investigated using Fourier transform infrared (FTIR) spectroscopy. For this purpose, a Bruker ALPHA FT-IR spectrometer with a platinum ATR was used.

The variations in the surface properties of the calcite particles resulting from the modification procedure were examined using scanning electron microscopy (SEM). The images were recorded using the FEI Quanta FEG 250 SEM. The morphological features of the polyDCPD composites were also investigated using SEM (ZEISS Supra 40 VP). For this purpose, each sample was first immersed in liquid nitrogen and then sectioned and mounted on copper grids. Before the SEM investigation, all samples were sputtered with gold under vacuum. Afterwards, the recorded SEM images were used to calculate the average cavity diameter (R1) and interconnected pore diameter (R2) of the samples. To determine the average cavity diameter, at least 80 measurements were taken from the SEM images of each sample using an image analysis software package (ImageJ, https://imagej.nih.gov/ij/download.html (accessed on 4 October 2022)), and the average value was then calculated. Afterwards, this value was corrected by multiplying with a factor (2/3^1/2^) [36]. The same method was used for the calculation of the average interconnected pore size: at least 100 measurements were taken from the SEM images of each sample, and average values were calculated and multiplied with the correction factor (2/3^1/2^). The mean diameters measured in this way are estimated below the real values as a reflection of the sectioning performed at a random distance from the voids of the samples and the center of the void. Multiplying the average value of the void diameters with this correction factor provides better estimations of the real diameters of the cavities and interconnected pores [36].

An N_2_ adsorption/desorption analysis was performed to determine the Brunauer–Emmet–Teller (BET) specific surface areas (R3) of the samples. With this aim, all samples were first subjected to a degassing procedure using the Micromeritics FlowPrep 060 Sample Degas System (Micromeritics Instrument Corporation, Norcross, GA, USA). Degassing was carried out at room temperature for 24 h. N_2_ adsorption/desorption isotherms were recorded using the Micromeritics Gemini VII 2390 t General Automated Surface Area and Porosity Analyzer (Micromeritics Instrument Corporation, USA) at 77.3 K. Surface area measurements utilized a nine-point adsorption isotherm collected over 0.05–0.20 P/P_o_. The foam density (R4) of the samples was calculated according to Archimedes’ principle using an analytical balance (Weightlab WSA224T) equipped with a density determination kit. For the determination of the R3 and R4 measurements, they were repeated three times using different specimens of each sample, and the arithmetic mean of the three measurements was calculated for each property.

The compressive moduli (R5) of the polyDCPD composites were determined using a ZwickRoell Z020 Universal Tester (Zwick GmbH&Co.KG, Ulm, Germany) equipped with a 10 kN load cell. For this purpose, cylindrical specimens with a diameter of ~25 mm and a height of ~10 mm were used. Uniaxial compression tests were performed according to ASTM D1621 (Standard Test Method for Compressive Properties of Rigid Cellular Plastics). The tests were repeated five times using five different specimens for each composite sample. The stress–strain plots were plotted using the data entered into testXpert II original software (ZwickGmbH&Co.KG, Germany). The compression moduli (R7) of the samples were also determined using testXpert II original software (ZwickGmbH&Co.KG, Germany). The R5 was calculated for each sample by taking the arithmetic average of the compressive modulus values determined from five different measurements of each sample.

### 2.4. Statistical Analysis

The experimental results were statistically analyzed using Minitab^®^ 21.1© 2021 (Minitab, LLC) statistical software for Windows. For this purpose, the data obtained from the material characterization were entered into the Minitab^®^ 21.1 Software according to the experimental design matrix given in Table 2. According to Table 2, the “Center Pt” column shows experiments for which the parameter levels are set in the center of the low and high settings. In this column, “1” represents the corner point while “0” denotes a center point. Moreover, the “Blocks” column indicates the groups of experiments which are conducted under the same conditions.

## 3. Results and Discussion

### 3.1. Experimental Results

The cryoscopic method is a simple approach used to achieve surface modification through physical interactions. The basic principle involves forming a suspension of particles to be surface modified with the modifier molecules, usually in an aqueous environment. During solidification, ice crystals are formed and grown in the suspension. Accordingly, the dispersed solid particles (calcite) and the modifier molecules in the suspension are rejected by the moving solidification front, concentrated, and trapped between the crystals. Once the solidification of the suspension is completed, the aqueous phase is sublimated at a low temperature and reduced pressure using freeze-drying equipment. The removal of the dispersing water phase is followed by the formation of hydrogen bonds between the modifier molecules and the solid particles [37]. Herein, hydrogen bonds are formed between the hydroxyl groups of the triblock copolymer and the calcite. The modification of the calcite particles was confirmed via FTIR. For this purpose, the FTIR spectra of calcite particles, Pluronic^®^ P-123, and *m*Calcite were compared (Figure 1). In Figure 1a, the peak at around 3600 cm^−1^ corresponds to the structural water. The peak at 711 cm^−1^ represents the symmetric CO_3_ vibrations of the crystalline calcium carbonate phases. Finally, the peaks arising at 1397 cm^−1^ and 871 cm^−1^ correspond to the vibrations of the carbonate ions [38]. In Figure 1b, the peak at 2869 cm^−1^ is due to the CH_2_ stretching vibrations in the triblock copolymer, while the peaks arising at 1372 cm^−1^ and 1457 cm^−1^ are due to the CH_3_ symmetric and antisymmetric deformation vibrations, respectively [39]. In addition, a C-O stretching band is observed at 1095 cm^−1^. As seen in Figure 1c, the spectrum of *m*Calcite displayed all the characteristic bands of both calcite particles and Pluronic^®^ P-123, while no new additional peaks attributable to chemical bonding were observed. The results confirmed that the physical modification of the calcite particles via the cryoscopic approach was successfully performed.

To examine the variations in the surface morphology of the particles, a SEM investigation was conducted before and after surface modification using Pluronic^®^ P-123. The SEM images of the calcite and *m*Calcite at different magnifications are presented in Figure 2. It can be seen clearly from the SEM images that the surfaces of the calcite particles were coated with the triblock PEG–PPG–PEG copolymer (Figure 2c,d) after surface modification.

The ROMP of the DCPD-based HIPEs was prepared using the experimental design matrix given in Table 2 and resulted in 12 polyDCPD composite samples. In each case, we did not observe phase separation during the polymerization of the HIPE templates. Moreover, all samples were in the form of monoliths. The hierarchical macroporous morphology of polyDCPD composites were investigated using SEM and the recorded images are presented in Figure 3 and Figure 4. Figure 3 and Figure 4 reveals that, in all cases, porous polyDCPD composites were obtained. Moreover, the resulting composite materials mostly exhibit the hierarchical open-cellular morphology of the emulsion-templated polymers. The variations in pore morphology of the obtained polyDCPD composites can be clearly seen from the SEM images. This variation can be attributed to the influence of the experimental parameters, namely, the surfactant amount, the *m*Calcite amount, and the internal phase ratio. Each parameter has an influence on pore morphology, though synergistic interactions are also important. However, the precise influence of changing the experimental parameters can be evaluated using a statistical analysis. In this respect, the difference between the SEM images can be identified by the evaluation of the R1 and R2 responses. In addition, it can be seen from Figure 4 that in some cases the cavity walls are deteriorated due to the deformation caused by the mechanical strength of the polyDCPD-based polymer matrix during the preparation of the samples for microscopy.

Depending on the SEM images, it was calculated that the average cavity diameter (R1) of the polyDCPD composites varied between 3.91 µm and 19.75 µm, whereas the interconnecting pore diameters (R2) varied between 1.15 µm and 5.82 µm (Table 3). Moreover, the average cavity diameters and interconnecting pore diameters of the control samples (9KP, 10-KP, 11-KP, and 12-KP) were found to be in good agreement, which can be considered an indication of the accuracy and reproducibility of the experiments. The BET specific surface areas (R3) of the polyDCPD composites were also determined from the N_2_ isotherms and tabulated in Table 3. The BET specific surface areas of the composites varied between 2.46 m^2^/g and 5.70 m^2^/g. These low surface area values are due to the macroporous structures of the composites and are coherent with the previously reported values of emulsion-templated macroporous polymers [2,31,33,34].

The foam density (R4) of the obtained composite materials was investigated using Archimedes’ principle. It was determined that the foam density of the polyDCPD composites varied between 0.27 g/cm^3^ and 0.63 g/cm^3^ (Table 3). Once again, note that the two samples exhibiting the lowest and highest densities (8-KP and 4-KP, respectively) were obtained by polymerizing HIPEs with different internal phase ratios (85 vol% for 8-KP and 75 vol% for 4-KP).

The mechanical strength of the polyDCPD composites was determined using compression tests performed at room temperature. The stress–strain curves that were recorded using the data obtained from the original software of the test device are presented in Figure 5, whereas the compressive moduli (R5) of the samples are tabulated in Table 3. It can be seen from Figure 5 that all the composite samples exhibit similar deformation behavior under compressive force. As with all the other measured properties, it was observed that the mechanical properties of the polyDCPD composites changed depending on the experimental parameters. On the other hand, the compressive modulus values of the control samples (9-KP, 10-KP, 11-KP, and 12-KP) were found to be close to each other, as expected (in the range of 31.3–37.9 MPa), and the highest value was recorded for 2-KP (130 MPa) (see Table 3). Sample 2-KP has the advantage of low nominal porosity (75 vol% of internal phase), leading to a low ratio of voids in the polymer matrix.

### 3.2. Statistical Analysis

The variation between the experimental results shows that the reciprocal and synergetic interactions of the experimental parameters should be taken into account, apart from the main effects, in the evaluation of the results. In this context, by combining the results from the experimental measurements and statistical analysis, the relationship between the experimental parameters presented in Table 1 (surfactant amount (A), mCalcite amount (B), and internal phase ratio (C)) and the properties of the polyDCPD composites were modeled. The influence of the experimental parameters on the average cavity diameter (R1), interconnecting pore diameter (R2), BET specific surface area (R3), foam density (R4), and compression modulus (R5) were examined using a statistical approach. The major and combined effects of the control factors were determined at a 95% confidence level (α = 0.05) using Minitab^®^ 21.1© 2021 (Minitab, LLC) statistical software. The influence of the experimental parameters on the selected responses (R1–R5) are expressed through the mathematical model equations based on the response surface design methodology given in Equation (1) and in Equations (2)–(6). In addition, regression coefficients, standard errors, t- and *p*-values, R-sq and R-sq (adj) values for each feature (all responses from R1 to R5) are also presented in Appendix A. The magnitude of the main effects and the combined effects of the experimental parameters on the responses (R1 to R5) was also determined by applying a Student’s *t*-test. Accordingly, Pareto charts for each response are also presented in Appendix A.
R = β0 + β1A + β2B + β3C + β4A*B + β5A*C + β6B*C + β7 A*B*C + β8CtPt(1)

In Equation (1), β0 represents the constant term as the global mean in the equation, while βi represents the regression coefficients corresponding to the main effects and combined effects. CtPt corresponds to the center points. A positive value of a factor indicates an increase in the response obtained when that factor changes from a low level (−1) to a high level (+1), whereas a negative value indicates that the numerical value of the response decreases.
R1 = 7.932 − 1.505A − 2.007B + 2.041C + 2.211A*B − 1.211A*C − 0.476B*C + 2.365A*B*C + 0.867CtPt (2)
R2 = 2.897 + 0.095A + 0.003B + 1.506C + 0.494A*B + 0.202A*C + 0.196B*C + 0.436A*B*C + 2.522CtPt(3)
R3 = 3.819 + 0.140A + 0.752 B + 0.292C − 0.468A*B − 0.107A*C − 0.138 B*C − 0.541A*B*C − 0.215CtPt(4)
R4 = 0.5367 − 0.0133A − 0.0113B − 0.0658C − 0.0561A*B − 0.0364A*C − 0.0330B*C - 0.0426A*B*C − 0.1699CtPt(5)
R5 = 56.16 − 9.29A − 20.41B − 36.61C − 1.16A*B − 3.56A*C − 10.31B*C + 16.06A*B*C − 22.01CtPt(6)

The influence of nanoparticle loading on the average cavity diameter and interconnecting pore size of the nanocomposite polyHIPEs was previously reported by Kovačič et al. [40]. According to their findings, preparing polyHIPEs by nanoparticle loading changes the average cavity diameter and interconnecting pore size. However, they did not explain how the main parameters and synergestic interactions influence these features. On the other hand, in our previous study [5], we reported the statistical importance of the binary effect of the internal phase ratio and the amount of nanoparticle loading on the average cavity diameter and average pore size of polyDCPD-based polyHIPE nanocomposites. Herein, we have also shown the significance of the triple interaction effect of the internal phase ratio, the surfactant amount, and the amount of nanoparticle loading. In this respect, Equation (2) shows the relation of the average cavity diameter (R1) in the polyDCPD composites with the selected parameters. Based on the main effects of the regression coefficients, the determination of R1 was found to change its order to C > B > A. The model equation demonstrates that the increase in the internal phase ratio (C) increases the R1, while the increase in the surfactant amount (A) and the amount of *m*Calcite loading (B) decrease this property. Considering the regression coefficients in Equation (2), it can be stated that binary (A*B) and triple effects (A*B*C) also have significance in terms of the R1. The contribution of the combined effects was found to change the order to A*B*C > A*B > A*C. This indicates that the average cavity diameter was mostly determined by the synergistic interaction of the experimental parameters. On the other hand, the Pareto chart for the R1 (Appendix A) and *p*-values (analysis of variance, ANOVA, Appendix A), presented in the SI document, reveal that all the combined effects have a statistically significant influence on the determination of the R1, except for the binary B*C effect with *p*-value > 0.05.

The regression coefficients in Equation (3) express the positive contribution of the main and combined effects on the variations in the interconnecting pore diameters (R2) of the polyDCPD composites. It can be seen from Equation (3) that the magnitude of the main effects changed the order to C > A > B, whereas the regression coefficient of B in Equation (3) shows that the main parameters of the amount of *m*Calcite loading (B) and the surfactant amount (A) have a negligible influence on the determination of the R2. These results can clearly be noticed from the *p*-values of (A) and (B) given in Appendix A, which are higher than 0.05. The regression coefficients demonstrated the importance of the combined effects for the determination of this property. In this respect, binary A*B and triple A*B*C effects make a statistically meaningful contribution. These findings were also supported by the Pareto chart for the R2 presented in Appendix A: C, A*B, and A*B*C were the main and combined effects with statistical significance.

In our previous work, we prepared polyDCPD-based polyHIPE nanocomposites using chemically modified CNCs [5]. Statistical analysis of the results showed that the surfactants did not have a significant effect on the R1 or R2 responses in these nanocomposites. This can be explained by the effective role of nanoparticles in emulsion stabilization. Unlike the previous work, here we prepared polyDCPD-based polyHIPEs using micronized calcite particles, which were modified using a cryoscopic approach. In the surface-modification process used, physical interactions were dominant between the particles and the modification agent. Consequently, the use of an additional surfactant has also become important, as the strength of the interactions plays an important role in the stabilization of the emulsion. This finding is evident from the equations derived for the R1 and R2 responses (Equations (2) and (3)), and the pareto charts (Appendix A) for these responses. The binary interaction effect of amount of surfactant (A) and nanoparticle amount (B) is statistically significant for both responses.

The effect of the amount of nanoparticle loading on variations in the BET specific surface area of the polyHIPE nanocomposites was also investigated by several groups previously. In this respect Kovačič et al. reported that nanoparticle loading did not have a significant influence on the surface area of the polystyrene-based polyHIPE nanocomposites [40]. In another study, Çetinkaya et al. reported that surface area of Mn_3_O_4_/p(DCPD)HIPE nanocomposites changes with the amount of nanoparticles [41]. In here, statistical analysis was conducted to explain how the amount of nanoparticle loading influences the surface area of polyHIPEs. Accordingly, contribution of the main effects and combined effects, which have statistical meaning in determination of BET specific surface area (R3) of polyDCPD composites is also expressed by R3 model equation given in Equation (4). All the main effects have positive contribution to R3 and significance level of their self contribution changes as follows: B > C > A. Based on the regression coefficients of binary and triple effects in Equation (4) the combined effects of the selected experimental parameters also have considerable importance and their synergistic effect leads decreament of R3. On the other hand, magnitude of combined effects can be listed as A*B*C > A*B > B*C and > A*C. These results indicates that the amount of *m*Calcite loading (B) has a great importance also within the combined effects. When these findings were compared to Pareto chart for the R3 (see Appendix A) and ANOVA Table of BET specific surface area (see Appendix A), it was concluded that the main contribution of B and triple contribution of A*B*C have statistically significant influence on R3 response. Compared to previously reported results, it could be concluded that our findings in this study are coherent with those in references [40,41] and confirm the statistical significance of the relation between the amount of nanoparticle loading and surface area of polyHIPEs.

The dependence of the foam density (R4) of the obtained polyDCPD composites on the experimental parameters was also investigated statistically, and the R4 model equation given in Equation (5) was derived. From Equation (5), the significance levels of main effects in determining the R4 can be expressed as C > A > B, while the order of the combined effects can be written as follows: A*B > A*B*C > A*C > B*C. Considering the regression coefficients and *p*-values presented in Appendix A, it can be also seen that the most important and statistically significant parameter among the main factors affecting the R4 is the internal phase ratio (C). Since the regression coefficients in Equation (5) are negative, both the main and combined effects have a decreasing influence on the foam density of the obtained composites. On the other hand, the Pareto chart for the R4 (see Appendix A) reveals that the R4 was determined by the contributions of C and A*B. The reason why the internal phase ratio is the most important parameter in determining the foam density is that the voids of the polyHIPEs are created by the removal of the internal phase [2,3,12]. This result is also consistent with our previous study [5].

According to the R5 model equation given in Equation (6), the compression modulus, which is a measure of the mechanical strength of the obtained polyDCPD composites, is strongly dependent on both main and combined effects. In this context, the order of importance of the main effects on the R5 equation is expressed as C > B > A, while the contribution levels of the binary and triple effects can be expressed as A*B*C > B*C > A*C > A*B. In addition, when the regression coefficients are examined, it is seen that all the main and combined effects, except for the triple effect A*B*C, contribute negatively to the R5. In addition, the Pareto graph (Appendix A) reveals that all the main and combined effects, except A*B (*p*-value > 0.05), make a statistically significant contribution to the R5 equation. The determination that the internal phase ratio is the most important parameter affecting the compressive modulus response is also consistent with the results obtained for the foam density, and is thus an expected result. It also supports the data we obtained in our previous study. As a result, as the internal phase ratio increases, the voids in the material increase, the foam density decreases, and the mechanical properties decrease.

## 4. Conclusions

Herein, a statistical approach was implemented to suggest a systematic method for the preparation of macroporous polyDCPD composites. Macroporous composites were synthesized via high internal phase emulsion (HIPE) templating, where emulsion formulations were created using an experimental design methodology. The experimental parameters were selected according to the dominant effects on the pore morphology and physical properties, such as average cavity diameter, interconnecting pore diameter, BET specific surface area, foam density, and compression modulus, which are referred to as responses R1, R2, R3, R4, and R5, respectively. The HIPEs of DCPD were stabilized using a combination of PEG–PPG–PEG-modified calcite particles and a non-ionic surfactant. In addition to the internal phase volume, the influence of the amount of surfactant and particle loading was studied in three distinctive levels based on the experimental design matrix. Accordingly, the effect of the emulsion formulation on the selected responses was revealed. 

The ANOVA results with a 95% confidence level (α = 0.05) showed that different main and interaction effects significantly contributed to the determination of the final specific composite properties. For the average cavity size and interconnecting pore size, it was found that A*B*C, A*B, and C were the most significant effects. In particular, the average cavity size was found to be mostly determined by the synergistic interaction of the surfactant amount (A), the amount of particle loading (B), and the internal phase ratio (C), while the interconnecting pore size was mainly dependent on the internal phase ratio (C). Moreover, variations in specific surface area were found to be influenced strongly by the amount of particle loading (B). The compression modulus and foam density, which are known to be important features in terms of mechanical properties, were also found to be influenced by the main effect of the internal phase ratio (C).

Overall, with this systematic investigation, macroporous polyDCPD/*m*Calcite composites were synthesized for the first time. It has been demonstrated that the morphological and mechanical properties of the composites can be altered by changing the experimental parameters. It has also been shown that *m*Calcite can be used as a filler to improve mechanical strength, reduce raw material costs, and increase material functionality for further applications. We believe that the resulting materials are good candidates for use as support material for obtaining shape-stabilized PCM composites. Therefore, this issue will be addressed in future studies.

## Figures and Tables

**Figure 1 polymers-15-00228-f001:**
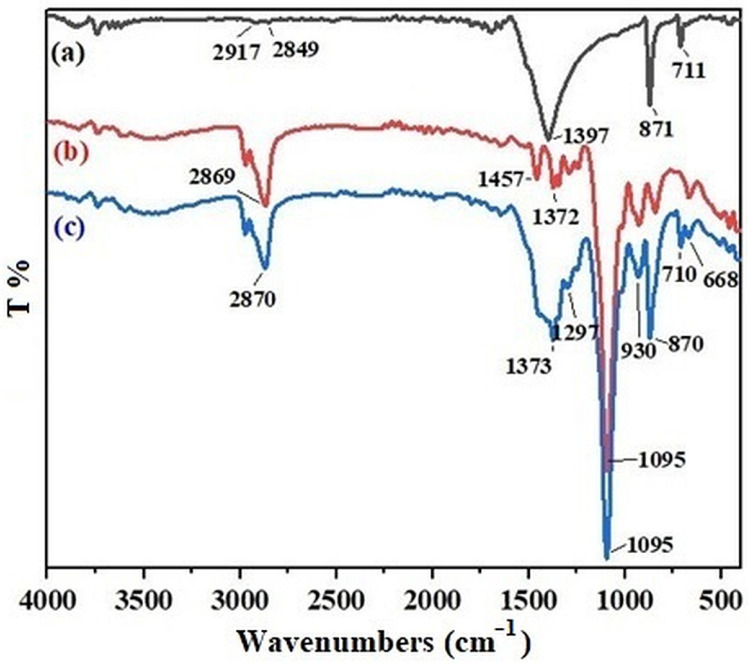
Comparative FT-IR spectra of (**a**) calcite, (**b**) Pluronic^®^ P-123, and (**c**) *m*Calcite.

**Figure 2 polymers-15-00228-f002:**
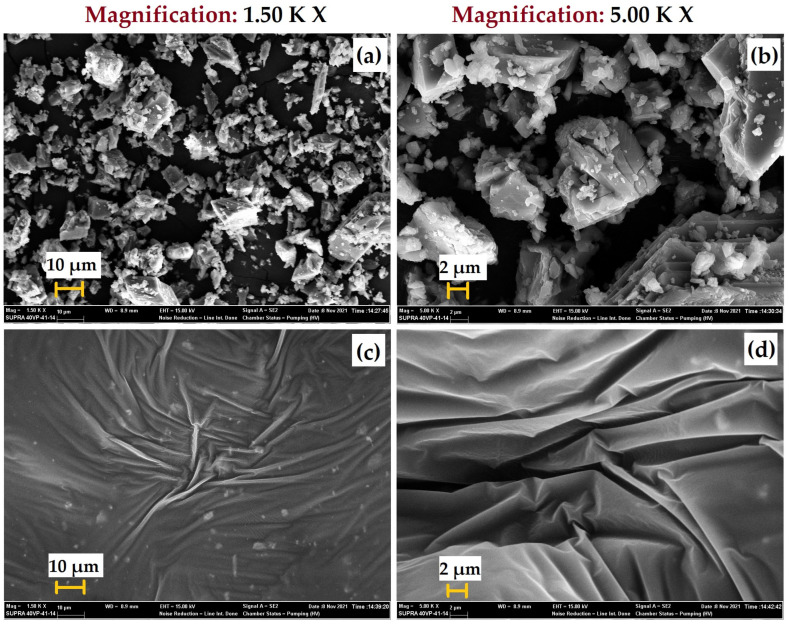
SEM images of (**a**), (**b**) calcite and (**c**), (**d**) *m*Calcite.

**Figure 3 polymers-15-00228-f003:**
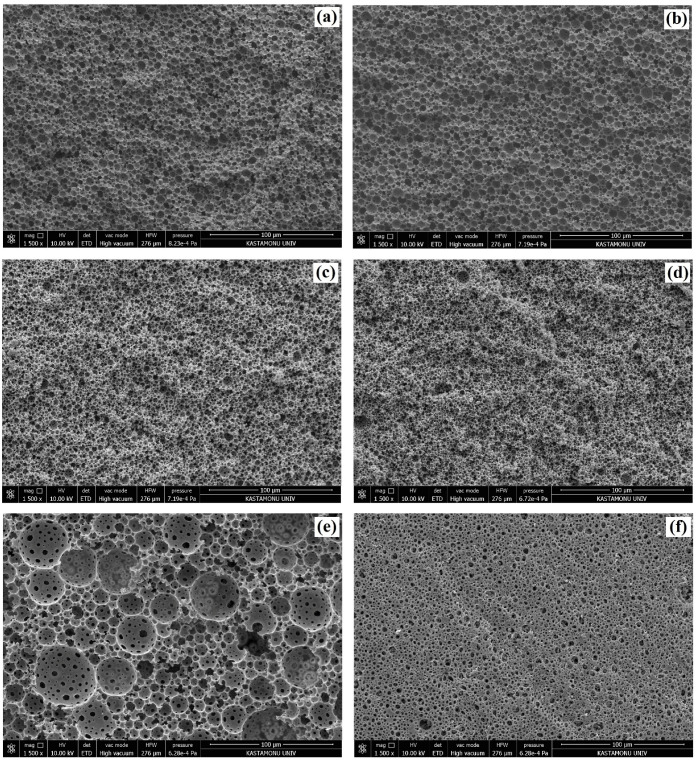
SEM images of polyDCPD composites: (**a**) 1-KP, (**b**) 2-KP, (**c**) 3-KP, (**d**) 4-KP, (**e**) 5-KP, and (**f**) 6-KP.

**Figure 4 polymers-15-00228-f004:**
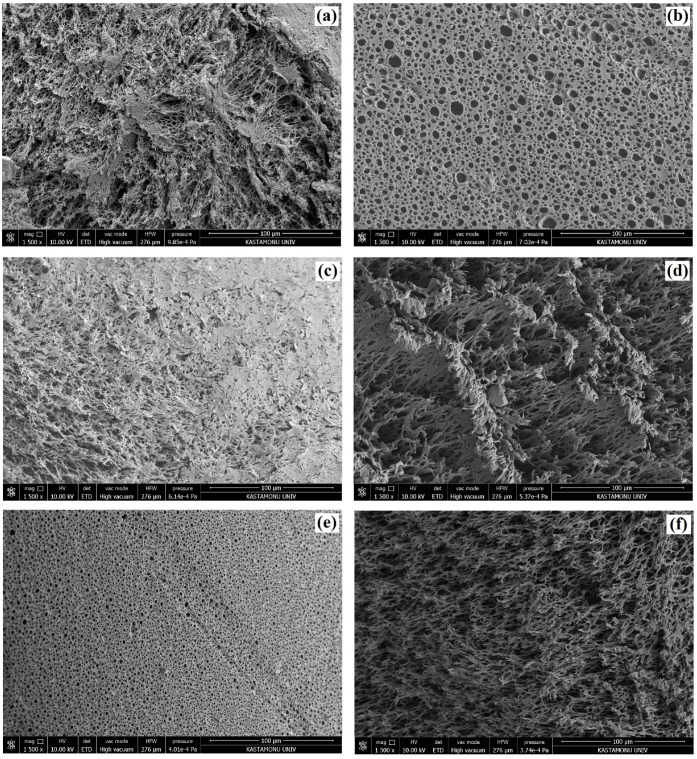
SEM images of polyDCPD composites: (**a**) 7-KP, (**b**) 8-KP, (**c**) 9-KP, (**d**) 10-KP, (**e**) 11-KP, and (**f**) 12-KP.

**Figure 5 polymers-15-00228-f005:**
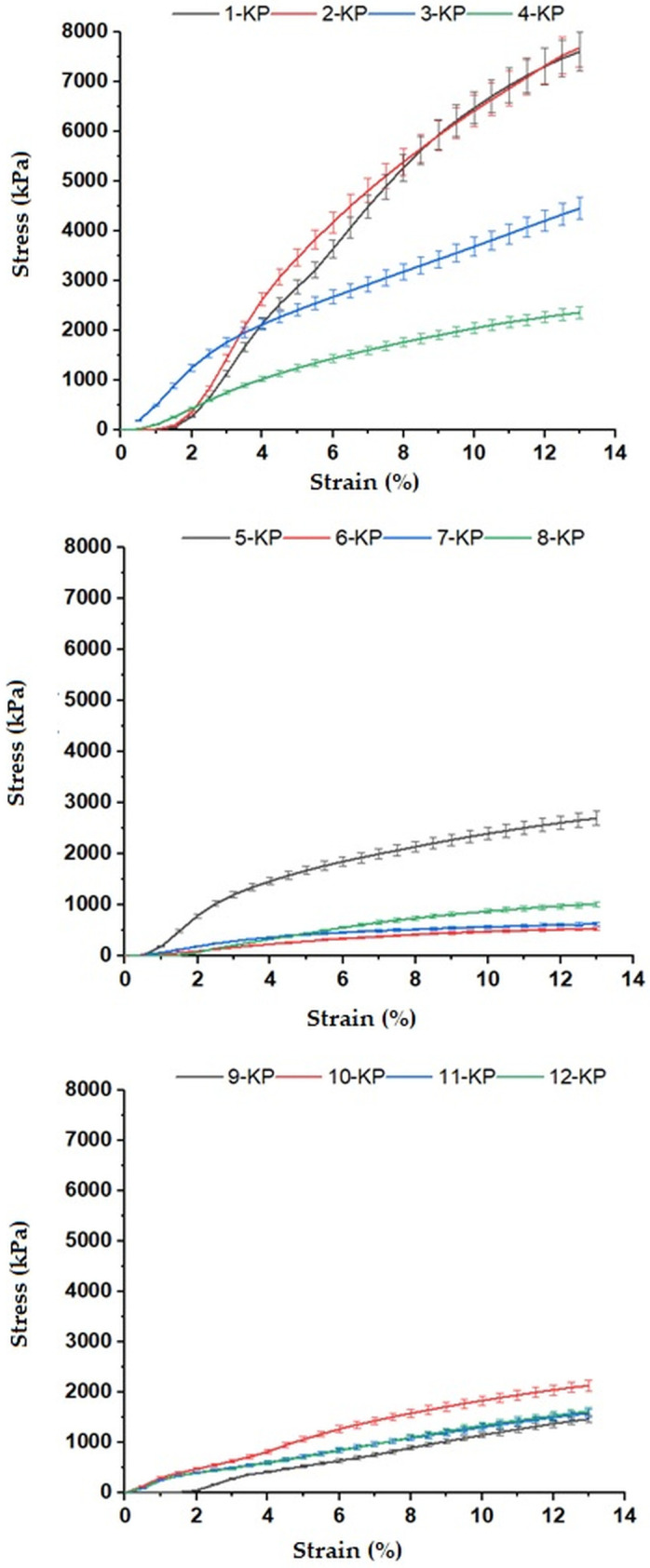
Stress–strain curves of polyDCPD composites.

**Table 1 polymers-15-00228-t001:** Experimental parameters and factor levels.

	(A)	(B)	(C)
Level	Surfactant Amount (vol%)	*m*Calcite Amount (wt%)	Internal Phase Ratio (vol%)
−1	3	1	75
0	4	3	80
1	5	5	85

**Table 2 polymers-15-00228-t002:** Experimental design matrix.

StdOrder	CenterPt	Blocks	Levels		
			(A)	(B)	(C)
1-KP	1	1	−1	−1	−1
2-KP	1	1	1	−1	−1
3-KP	1	1	−1	1	−1
4-KP	1	1	1	1	−1
5-KP	1	1	−1	−1	1
6-KP	1	1	1	−1	1
7-KP	1	1	−1	1	1
8-KP	1	1	1	1	1
9-KP	0	1	0	0	0
10-KP	0	1	0	0	0
11-KP	0	1	0	0	0
12-KP	0	1	0	0	0

**Table 3 polymers-15-00228-t003:** Characterization results for polyDCPD composites (R1: average cavity diameter; R2: interconnecting pore diameter; R3: BET specific surface area; R4: foam density; R5: compression modulus).

Sample	R1 (µm)	R2 (µm)	R3 (m^2^/g)	R4 (g/cm^3^)	R5 (MPa)
1-KP	7.56 ± 0.17	1.75 ± 0.03	2.46 ± 0.02	0.54 ± 0.01	107.13 ± 4.22
2-KP	7.28 ± 0.20	1.41 ± 0.03	2.81 ± 0.02	0.61 ± 0.04	130.02 ± 5.16
3-KP	4.81 ± 0.14	1.25 ± 0.02	4.10 ± 0.21	0.61 ± 0.05	80.25 ± 3.06
4-KP	3.91 ± 0.14	1.15 ± 0.03	4.74 ± 0.27	0.63 ±0.06	34.22 ± 1.73
5-KP	19.75 ± 1.49	4.83 ± 0.14	2.46 ± 0.03	0.46 ± 0.01	62.17 ± 2.87
6-KP	5.16 ± 0.19	3.57 ± 0.15	4.53 ± 0.25	0.56 ± 0.03	7.08 ± 0.36
7-KP	5.63 ± 0.23	3.37 ± 0.11	5.70 ± 0.32	0.57 ± 0.04	12.27 ± 0.81
8-KP	9.35 ± 0.29	5.82 ± 0.20	3.75 ± 0.18	0.27 ± 0.01	16.19 ± 0.95
9-KP	8.26 ± 0.15	5.53 ± 0.09	4.07 ± 0.18	0.33 ± 0.02	35.14 ± 1.03
10-KP	8.89 ± 0.27	5.06 ± 0.16	3.29 ± 0.26	0.32 ± 0.01	38.34 ± 1.55
11-KP	8.23 ± 0.11	5.31 ± 0.07	3.86 ± 0.42	0.42 ± 0.04	31.86 ± 1.37
12-KP	9.81 ± 0.27	5.78 ± 0.21	3.19 ± 0.31	0.39 ± 0.02	30.97 ± 1.69

## Data Availability

Not applicable.

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
