# Peer review of "Hierarchical Macroporous PolyDCPD Composites from Surface-Modified Calcite-Stabilized High Internal Phase Emulsions"

_polymers, 2023, doi:10.3390/polym15010228_

Round 1
Reviewer 1 Report
The work by Eslek et al. looks at a potentially interesting and industrially relevant system of microporous polydicyclopentadiene mixed with micron sized calcite. By my assessment, the topic is of interest to the readership of Polymers, and systematic studies such as these are always a welcomed addition to the field. However, there are a number of open questions or important details missing that need to be resolved so that the submitted manuscript can be accurately evaluated for its scientific merit. I expand on a number of these points below. I am not able to recommend the article for publication in its current form; however, I would like to comment that I believe the authors’ approach to the system at hand can be appropriate with proper methodology.
Comments:
Minor point: In the introduction, the authors use the phrase “real industrial applications.” The word real here is unnecessary and should be removed. This implies that there are somehow “fake industrial applications.”
Minor point: Also in the introduction, “Since macropores are dominated the pore morphology, polyHIPEs usually exhibits low-mechanical strength.” I believe the authors mean “Since macroporous material features are dominated by the pore morphology, polyHIPEs usually exhibit low-mechanical strength.”
Major point: In line with the last comment, there are several grammar errors or incomplete sentences throughout the manuscript that need to be resolved before acceptance. A careful re-read and editing is required.
Minor point: In the materials section, the authors list the exact purchasing information of all of their materials, including the triblock copolymer. This is an appreciated standard practice, however, they should also list the sequence length of each block. This information helps with interpreting the final system.
Major point: One significant deficit of the presented work is the lack of characterization and surrounding discussion of the micronized calcite. Even the preparation method is missing. The only information given is that the particles were provided by their colleague. This significantly reduces the interpretability of the manuscript.
Minor point: The section of modification of Calcite should provide some description of how cyroscopic preparation modifies the surface. For example, a brief discussion of the attachment mechanism would be an aid to the reader. This will also enhance the point the authors make associated with the FTIR spectra.
Minor point: The Table 2 caption is lacking details. “Experimental design matrix” is vague. As an example, CenterPt and Blocks should be defined (either as a footer of part of the caption depending on the journal format). I am aware that these are defined later down in the work, but the reader sees this information long before it is discussed.
Minor point: In the sample characterization section, the authors mention a correction factor of 2/3^(1/2). Context should be given for why this factor is used/why it is needed/where it comes from.
Major point: Continuing with sample characterization, assessment of pore size from 80 also needs to be discussed. Arguably, this is the largest deficit in the manuscript. Examples, are the 80 measurement randomly chosen? Does this give a representative PSD? How were pores within a sample image chosen: was this also randomly? Was this done by visual inspection of the researcher? Was only 1 image chosen for 80 measurements or were 80 measurements take across multiple micrographs? Conceivably more images would be more representative of the distribution of pores each sample possesses.
Major point: Continuing with sample characterization, the authors need to provide additi9onal details on how they calculated interconnected pore diameters. It is not clear from the text how this information was extracted from the SEM micrographs.
Major point: Numerous details are missing regarding the nitrogen sorption isotherms used for BET analysis. At a minimum the authors need to discuss the exposure time at each adsorption state point. Moreover, there are other details, such as how the isotherm region for BET analysis was chosen. I am not able to judge the surface area analysis presented without these details.
Major point: In the SEM images of Figure 3, the sample with deteriorated walls should not be included in the following analysis. These samples are significantly different than the rest, and therefore, presenting them as an apples-to-apples grouping in all of the following analysis is misleading. The selection of “what is a pore” in these samples is subject to significant bias of the researcher.
Major point: In Table 3, all values need to be reported with measures of uncertainty. If systems and measurements are performed in replicate, which they should be, then it should be possible to assign a 95% confidence interval to all values in this table. This is necessary to judge statistical significance.
Uncertain point: I believe it is not possible to evaluate the conclusions and discussion in Section 3.2 without the above points being remedied. The fitting and statistical analysis clearly depends on how the reference values were obtained. The analysis, discussion, and conclusions will need to be revisited after addressing my previous comments
Author Response
Dear reviewers,
We are grateful for the good suggestions and valuable comments about our work. All the comments are very helpful to improve our paper. We have tried our best to address the referees′ comments one by one. All revised contents with red color are presented in the revised manuscript. Other explanation and corresponding supporting information can be found in the attached files.
Best regards,
Mohamed Alaasar.

Reviewer 2 Report
The paper presents the ring opening metathesis polymerization of HIPE-DCPD and the variations of morphological and physical properties of the synthesized composites as a function of the surfactant amount, mCalcite amount, and the internal phase ratio. The results, especially the statistical analysis is very interesting in terms of analyzing the polymerization in terms of different variables, however, the manuscript must be revised to improve its readability and to address the following issues concerning the presentation and analysis of the data:
1. All abbreviations, symbols, and variables must be defined when first used in the text. While most of them were defined, some were missed by the authors, one example is the BHT in section 2.1.
2. The donated micronized calcite was definitely purchased somewhere, it is good to indicate also where the donor purchased the substance.
3. For StdOrder 9-12, are they really all similar? Since in Figure 3, the results are different from each of them? How can you explain this when the reason for 9-12 being having similar parameters is to be used as control samples.
4. Table 2, rename StdOrder to be consistent with the remainder of the text, example, 1 as 1-KP.
5. In table 1, were the parameters determined based on existing literature?
6. Figure 1, are the FTIR spectra normalized? Is the y-axis relative T%? In the text, the broad water signature at 3600 cm-1 is not well observed.
7. Comparing figures 1 and 2, it seems that the calcite may only be covered by the Pluronic P-123 since the FTIR of mCalcite resembles the FTIR of Pluronic P-123. Do you have any other characterizations to show how the mCalcite was modified? The white spots on Figure 2 c and d might be the calcite traces that were detected in the 1373 acm-1 peak?
8. Figure 3 is too big, it covers 4 pages, are they all necessary to show and do the authors really need to show two types of magnifications? Only f, h-l were highligted in the text. Also, the SEM should tell a story, how does the surface change in terms of the parameters? Please elaborate in the text.
9. Table 3, how significant are the values shown in the table as compared to other samples? For example, in R4, how significant is 0.1? What are the uncertainties for each value?
10. Figure 4, what is the reason for splitting the data into 3 graphs?
11. The statistical analysis showed a good analysis of the data points extracted from the different characterizations, however, the discussion does not explicitly explained how changing each parameter affected the polymerization? An in-depth scientific discussion must be provided.
Author Response

(The authors gave the same response as above.)

Round 2
Reviewer 1 Report
The changes provided by the authors are sufficient. I maintain some procedural reservations, but I believe these are with standard practices in the field and are not specific to the submitted manuscript: i.e., they should not inhibit publication of the submitted work. I suggest the article be published. The additional details provided by the authors have appreciably improved the readability and scientific soundness of the manuscript. I have no further comments.
Reviewer 2 Report
After a careful review of the revised manuscript, I recommend to accept the manuscript for publication in Polymers. All my comments were addressed sufficiently.